# Pre-Treatment, Extraction Solvent, and Color Stability of Anthocyanins from Purple Sweetpotato

**DOI:** 10.3390/foods13060833

**Published:** 2024-03-08

**Authors:** Zhuo Chen, Jian Wang, Yang Lu, Qiang Wu, Yi Liu, Yonghua Liu, Sunjeet Kumar, Guopeng Zhu, Zhixin Zhu

**Affiliations:** 1School of Breeding and Multiplication (Sanya Institute of Breeding and Multiplication), Hainan University, Sanya 572025, China; chenzhuohd@163.com (Z.C.); wangjianhd@163.com (J.W.); lyanghd@163.com (Y.L.); wuqiangccb@163.com (Q.W.); 996178@hainanu.edu.cn (Y.L.); lyhjacky520@hotmail.com (Y.L.); 184224@hainanu.edu.cn (S.K.); zhuguopeng@hainanu.edu.cn (G.Z.); 2Key Laboratory for Quality Regulation of Tropical Horticultural Crops of Hainan Province, School of Tropical Agriculture and Forestry, Hainan University, Haikou 570228, China

**Keywords:** purple sweetpotato, anthocyanins, browning, pre-treatment, solvents, color stability

## Abstract

Purple sweetpotato anthocyanins (PSPA) exhibit significant potential as food colorants with associated health benefits. However, challenges related to browning and instability have hindered the application of PSPA. In this study, various pre-treatments and solvents for PSPA extraction were evaluated based on color, anthocyanin yields, antioxidant capabilities, and brown index. Browning markedly influenced the color and reduced the antioxidant capacity. Optimal results were obtained with the pre-treatment of “steaming of unpeeled whole sweetpotato” and the solvent “1% citric acid-ddH_2_O”. Furthermore, the color stability of purified PSPA solutions was evaluated under pH levels from 1 to 13 at 25 °C and 65 °C. The PSPA solutions showed a color spectrum from magenta, blue/green, and then to yellow across the pH range. The blue/green hues at pH 10–12 rapidly degraded, while the magenta hue at lower pH showed higher color stability. Elevated temperatures significantly accelerated the PSPA degradation. However, PSPA solutions at pH 1–2 exhibited remarkable color stability, with no spectral decay at either 65 °C for 12 h or 25 °C for 32 days. These results provide valid guidance for the extraction, preservation, and application of PSPA in the food industry.

## 1. Introduction

Anthocyanins are widely recognized as among the most potent plant antioxidants, with numerous health effects including prevention of cardiovascular diseases, anti-tumor, anti-aging, and anti-cancer properties [1,2]. With the concept of “coloring food with food”, many anthocyanin-rich fruits and vegetables are used to develop colorful foods with associated health benefits [3,4]. However, anthocyanins are susceptible to degradation and oxidation, leading to discoloration, browning, and decline of antioxidant activity during processing and storage [3,4,5,6]. These challenges related to anthocyanin stability hinder their application as food colorants or nutrients.

Sweetpotato (*Ipomoea batatas* L.) is one of the important food crops in Asia, and many purple varieties are known for their high content of purple sweetpotato anthocyanins (PSPA) [7]. PSPA has been identified mainly as glycosylation and acylation derivatives of cyanidin (Cy) and peonidin (Pn), and occasionally of pelargonidin (Pg) (Figure 1) [6,8,9,10]. The glycosylation pattern, specifically “3-sophoroside-5-glucoside” with three glucose molecules, is relatively fixed. While the acylation patterns vary, mainly found as mono- or di-acylated forms of *p*-hydroxybenzoic acid, caffeic acid and ferulic acid. Acylated anthocyanins were more stable than unacylated anthocyanins [11,12]. The very stable blue anthocyanins from the butterfly pea flowers are highly acylated [13]. And the acylated PSPA were proved to be more stable than anthocyanins from many other species, including grapes, red cabbages, and eggplants [13]. Furthermore, compared to many other plant sources of anthocyanins, purple sweetpotato offers higher yields at lower costs and easier accessibility [7]. Therefore, it is theoretically a good source of anthocyanin-based food colorants.

However, in addition to their specific structures, the stability of anthocyanins is greatly influenced by internal enzymes within plant tissues and external environment, including heat, pH, light, and O_2_ [6,12,14]. Polyphenol oxidase (PPO) induces browning by accelerating the oxidation of phenolic hydroxyls, and glycosidase can cause degradation by breaking the glycoside bonds in anthocyanin molecules [6,15]. Problems of browning and degradation, leading to discoloration and reduction of antioxidant activity, are significant obstacles to applying PSPA [6,16,17]. Hence, it is important to explore the extraction processes and preservation conditions that would minimize browning or degradation of PSPA.

The impact of extraction conditions, including heating pre-treatment or extraction solvent, on various properties of PSPA has been explored [8,17,18,19,20,21,22], but the conclusions about effectiveness vary. The influence of browning and subsequent long-term color stability needs to be taken into account. In this study, we investigated the PSPA extraction process using the commercialized purple sweetpotato cultivar “Violet”. This investigation covered the pre-treatments for sweetpotato, and the selection of solvents for extraction. Then, with purified PSPA, we evaluated the impact of pH and temperature on the properties of the PSPA solutions, including color stability, absorption spectra, and thermal degradation kinetics during storage. Varied browning during sweetpotato preparation were shown. The provided kinetics of color changes at a wider pH range and longer time span would be helpful in food design and preparations. These results provide valuable references for applications of PSPA in the food industry.

## 2. Materials and Methods

### 2.1. Materials

The purple sweetpotato “Violet” used in this study was a commercially available variety purchased online from Yulin, Guangxi Province, China. Raw sweetpotatoes were washed before various pre-treatments for the extraction of anthocyanins, including (**1**) **Fresh**; sweetpotatoes were peeled, shredded, and used immediately. (**2**) **Dried**; peeled and sliced sweetpotatoes were immediately dried in an oven at 60 °C for 10 h and then ground to powder. (**3**) **SD-s,** Steamed and dried of sliced sweetpotatoes; peeled and sliced samples were steamed at 121 °C for 10 min in the sterilization pot and then dried and ground as (2). (**4**) **SD-p**, Steamed and dried of peeled sweetpotatoes; tubers of sweetpotato were peeled and then steamed and dried as (3). (**5**) **SD-w**, Steamed and dried of whole sweetpotatoes; unpeeled whole tubers were steamed and dried as (3).

For samples exposed to drying, the dryness rate of 35 ± 0.65% was obtained by the ratio of dry weight to fresh weight. All analyses were conducted in triplicate, and data were shown as the means and standard error (SE).

### 2.2. Measurements of Color, Anthocyanins, Phenols, Flavonoids, and Antioxidant Capabilities for Samples by Various Pre-Treatments

#### 2.2.1. Measurements of Color

Objective color parameters by CIEL*a*b* were measured for sweetpotato slices exposed in the air and powders obtained from various pre-treatments, with a portable intelligent chromatic meter (LS170, LinShang Technology Co., Ltd., Shenzhen, China). The colorimeter has a built-in calibrating whiteboard on the cap. The light source is the full spectrum LED (D65) with a field view at 10°, and the measurements were done using the illumination method 45/0 (45° circular illumination, 0° reception). L* indicates brightness (black to white from 0 to 100); a* indicates greenish (−) and reddish (+); and b* indicates bluish (−) and yellowish (+). C (chromacity) shows the color intensity by the formula (a*^2^ + b*^2^)^0.5^. ΔE (chromatic aberration) shows the color difference between samples by (ΔL*^2^ + Δa*^2^ + Δb*^2^)^0.5^. Three replicates were measured.

#### 2.2.2. Measurements of Anthocyanins

Anthocyanins are extracted by 1% HCl-methanol and then measured by the pH differential method [23]. About 1 g fresh sample or 0.35 g dry powder (based on the specific dry rate for each replicate) was ground and homogenized in 10 mL 1% HCl-methanol. Supernatants were taken after centrifugation at 15,285× *g* (12,000 rpm) for 1 min (H1850R, Xiangyi Laboratory Instrument Development Co., Ltd., Xiangtan City, Hunan, China).

Dilution factors (DF) were recorded by diluting part of extracts with potassium chloride buffer (pH = 1) so A_530_ were around 1. According to the DF, extracts were mixed with potassium chloride buffer (pH 1.0) and sodium acetate buffer (pH 4.5), respectively. After 15 min equilibration at room temperature in the dark, A_530_ and A_700_ were collected (UV-2100, Unico (Shanghai) Instrument Co., Ltd., Shanghai, China) with ddH_2_O as blank, and anthocyanin content was calculated.
Anthocyanin content (mg/100 g FW) = 100 × (A × MW × DF × 1000 × V)/(ε × 1 × M)(1)

A = (A_530_ − A_700_) _pH 1.0_ − (A_530_ − A_700_) _pH 4.5_; MW: molecular weight of cyanidin-3-glucoside (449.2); DF: dilution factor; V: extraction volume; ε: molar absorption coefficient (26,900 for cyanidin-3-glucoside); 1: cuvette path length (1 cm); and M: sample mass calibrated as fresh weight. Three biological replicates were measured.

#### 2.2.3. Measurements of Phenols, Flavonoids, and Antioxidant Capabilities

The measurements were according to Shi et al. [24]. About 1 g fresh sample or 0.35 g dry powder was homogenized with 20 mL of 75% ethanol. Supernatants were collected after centrifugation (15,285× *g*, 1 min) for subsequent phenols, flavonoids, and antioxidant activity measurements. Data from three biological replicates were collected.

The total phenolic contents (TPC) were measured by Folin–Ciocalteu reagent (Guangzhou Chemical Reagent Factory, Guangzhou, China), with gallic acid as standard. TPC was evaluated by A_760_ and expressed as milligrams of gallic acid equivalents per 100 g fresh weight (mg GAE/100 g FW).

The total flavonoid contents (TFC, including flavones, isoflavones, flavonols, and dihydroflavonols) were measured with rutin as standard by Li et al. [25]. TFC was evaluated by A_508_ and expressed as milligrams of rutin equivalents per 100 g fresh weight (mg rutin/100 g FW).

Antioxidant activities were measured using trolox (i.e., water-soluble V_E_) as standard, by indices of 2,2′-diphenyl-1-picrylhydrazyl (DPPH) and 2,2′-azino-bis (3-ethylben-zothiazoline-6-sulphonicacid) (ABTS) free radical scavenging abilities. DPPH is hydrophobic, and its reactions must be run in organic solvents, such as anhydrous ethanol. In contrast, ABTS is water-soluble and can reflect antioxidant capacity in aqueous environments [26].

### 2.3. Properties of Anthocyanin-Based Extracts from Purple Sweetpotato by Four Solvents

#### 2.3.1. Selection of Solvents and Extraction of Anthocyanin from Purple Sweetpotato

The solvents were selected considering the edibility that can be applied to the food industry. Four solvents were compared: (**1**) **1% HCl-methanol**, an effective but inedible anthocyanin extractant in the laboratory, used here for comparison with other solvents. (**2**) **1% citric acid-75% ethanol**, that can be used in the wine industry. (**3**) **ddH_2_O** and (**4**) **1% citric acid-ddH_2_O**, that can be applied to the conventional food industry.

About 20 g fresh sample or 7 g dry powder was homogenized in 200 mL solvent. For solvent (1), the homogenate was left at room temperature for 1 h before later steps. For solvent (2–4), the extraction procedure given by Chen et al. was followed [22]. Briefly, the mixture was extracted in the water bath (80 °C, 40 min, and 100 rpm of shaking) and cooled rapidly to room temperature with tap water. After centrifugation at 10,610× *g* (10,000 rpm) for 10 min, supernatants were taken for subsequent usage.

#### 2.3.2. pH and Spectral Analysis of Extracts from Four Solvents

Values of pH for initial extracts were measured by pH meter (PB-30, Sartorius (Beijing) Instrument Co., Ltd, Beijing, China). In order to obtain solutions with theoretically identical concentrations of anthocyanins, the extracts were diluted with corresponding solvents by DF determined as Section 2.2.2. Continuous spectral scanning was performed from 380 to 700 nm (Infinite 200Pro, Tecan i-control, Männedorf, Switzerland). λmax for anthocyanins were obtained based on the absorption peaks.

#### 2.3.3. Determination of Anthocyanin Yields, Polymeric Color and Brown Index

Anthocyanin yields were measured by the pH differential method as in Section 2.2.2 [23]. Polymeric color and brown index were measured using the bisulfite bleaching method [23], in which monomer anthocyanins will combine with bisulfite to form colorless sulfonic acid adduct, while polymeric anthocyanins will not. Extracts from four solvents were diluted with ddH_2_O according to their respective DF, determined as Section 2.2.2. Following dilution, two aliquots of 2.8 mL diluted solution were added with 0.2 mL bisulfite solution and 0.2 mL ddH_2_O, respectively. After 15 min of equilibration, A_420_, A_530_, and A_700_ were collected for each sample with ddH_2_O as blank.

The A_420_ and A_530_ of the bisulfite-treated samples serve as indices for browning.
Brown index = A_420_/A_530_(2)

Color density was obtained by Formula (3) with A_420,_ A_530_ and A_700_ of the ddH_2_O mixed sample. Polymeric color was given by Formula (4) with A_420,_ A_530_ and A_700_ of bisulfite mixed sample.
Color density = [(A_420_ − A_700_) + (A_530_ − A_700_)] × DF(3)
Polymeric color = [(A_420_ − A_700_) + (A_530_ − A_700_)] × DF(4)
Polymeric color percent = (polymeric color/color density) × 100%(5)

### 2.4. Stability Analysis of Purple Sweetpotato Anthocyanin Solutions at Different pHs

#### 2.4.1. Extraction and Purification of Purple Sweetpotato Anthocyanins

With solvent (4), PSPA were extracted as Section 2.3.1. Supernatants were further purified by protocol of Yang et al. [27] with some modifications. Extracts were loaded onto an equilibrated AB-8 macroporous resin column at 5-bed volumes (BV)/h. The resin was washed with ddH_2_O at 4 BV/h to remove sugar and other impurities, and then eluted with 70% (*v/v*) ethanol. The ethanol eluent was collected by large-mouthed jars and blown in a clean bench for 24 h to get a thick syrup of anthocyanins.

#### 2.4.2. Stability Analysis of Anthocyanin Solutions at Different pHs

Citrate-phosphate buffer (pH 1.0 to 7.0), phosphate buffer (pH 8.0), and glycine-NaOH buffer (pH 9.0 to 13.0) were prepared by protocol given by Wu et al. [28]. High-concentration anthocyanin stock solutions were derived by dissolving the syrup with small volumes of 1% citric acid-ddH_2_O. According to the DF determined as Section 2.2.2, the stock solution was further diluted by buffers from pH 1 to 13 to obtain a series of pH solutions with the same concentration of initial anthocyanins.

Stability was explored in the dark under 25 °C or 65 °C, representing room temperature and pasteurization temperature, respectively. At time intervals of 3 days (at 25 °C) or 1 h (at 65 °C), aliquots of 200 μL solutions were taken into microplates, photographed, and subjected to continuous spectral scanning at 380–700 nm (Infinite 200Pro, Tecan i-control).

#### 2.4.3. Thermal Degradation Kinetics

λmax for anthocyanin was determined by the spectra, and thermal degradation kinetics were analyzed by absorption at λmax as Kirca et al. [29]. The degradation rate constant (k) and half-life time (t_1/2_) were calculated with Equations (6) and (7).
ln(A_t_/A_0_) = −k × t,(6)
t_1/2_ = −ln0.5 × k^−1^,(7)

A_0_ is the initial absorbance, and A_t_ is the absorbance at the time t. A higher k value means faster degradation, while a longer t_1/2_ means better stability.

## 3. Results

### 3.1. Effects of Pre-Treatments on the Anthocyanin-Related Properties of Purple Sweetpotato

From the appearance of newly cut slices of purple sweetpotato from 0 s to 20 min, the browning process happened instantly after exposure to air (Figure 2A). Powders derived from the oven “Dried” group showed a relatively fixed pink color. While powder from the “Steamed and Dried” group showed varied color depending on the browning extent during steaming (Appendix A and Figure 2B). SD-p was highly brown, SD-w presented a vibrant purple color. Objective color differences were shown by the CIEL*a*b* parameters (Figure 2C). For fresh slices, “F 0 s” had similar L*, a*, and b* as other reports [19,20], while the instant color change measurement was novel for this study: Decreased a* (from 24.5 to 11.1) and increased b* (from −4.2 to 11.1) indicated declining red hue and elevating yellow hue by browning process. The brown SD-s powder had the highest b* value (15.9). For fresh slices within 5 min, L* remained stable (around 27), C decreased gradually (from 18.2 to 9.9), and ΔE increased gradually (from 5.9 to 12.9). High C values were displayed for powder samples of various color (from 16.1 to 19.5). Compared to fresh slices at 0 s, ΔE were the highest for the pink D powder (28.5) and the brown SD-s powder (24.9). The lowest ΔE was displayed by the purple SD-w powder (3.1).

Contents for important indicators of the antioxidant capacities were determined, including anthocyanins, total phenolics and total flavonoids (Figure 2D). Two antioxidant indices were assessed, including the scavenging of DPPH and ABTS (Figure 2E). The five indices showed similar trends among samples. Fresh slices did not show any significant differences within 20 min, possibly because browning had not yet occurred in the inner flesh. Anthocyanin contents in fresh slices were highest (~150 mg/100 g FW), followed by SD-w (104.4 mg/100 g FW), SD-p (48.4 mg/100 g FW), D (40.5 mg/100 g FW), and SD-s (12.1 mg/100 g FW), respectively. PSPA yields reported in other studies [19,22] vary within this wide range, possibly due to different sweetpotato varieties or extraction methods. PSPA yields of fresh purple sweetpotato “TN57” are from 17.1 to 93.6 mg/100 g FW when various extraction factors are adopted [22]. The contents of total phenolics, total flavonoids, and antioxidant capacities exhibited similar trends. Among the four powder sources, SD-w had the highest antioxidant-related indices, and its corresponding pre-treatment of “steaming and oven drying of unpeeled whole sweetpotato” was indicated as optimal.

### 3.2. Optimal Solvents for the Anthocyanin Extraction of Purple Sweetpotato

The anthocyanin-related properties for extracts by four solvents were determined for samples of fresh sweetpotato and powders of “Dried” and “SD-w” (Table 1, Figure 3). Overall, extracts by four solvents differed significantly; properties for extracts from “Fresh” and “Dried” were similar, whereas they differed from that of “SD-w”.

The pH for initial extracts by the four solvents were around 0, 3.9, 6.0, and 2.4, respectively (Table 1). For the diluted solutions with theoretically identical PSPA concentration (Figure 3), magenta color and λmax around 526 nm were shown for solvent (1), (2), and (4), and the colors were light for solvent (2). The solutions of solvent (3) were brown for “Fresh” and “Dried” samples with no absorption peak identified, while they were pink for “SD-w” samples with λmax at 554 nm. No noticeable color decay was observed for solutions of solvent (1), (3), and (4) after 54 days’ storage. While for extracts from solvent (2), the light magenta color faded away completely by Day 25.

From anthocyanin yields calibrated as fresh weight (FW) (Table 1), “SD-w” samples were most productive. On aspects of solvents, solvent (1) was most effective, with similar yields of around 114 mg/100 g FW for “Fresh” and “SD-w” samples, and 41 mg/100 g FW for “Dried” samples. Solvent (4) ranked second with yields of around 80, 35, and 108 mg/100 g FW for “Fresh”, “Dried” and “SD-w” samples, respectively. The brown solutions of “Fresh” and “Dried” samples by solvent (3) showed the highest brown index of 7.86 and 2.33, and displayed polymeric color percent as high as 99.8% and 94.1%, respectively. The solution of “SD-w” samples with solvent (4) showed the lowest polymeric color percent (6.8%) and brown index (1.26).

Summarized from multiple indices, the “SD-w” samples extracted using solvent (4) were optimal.

### 3.3. Stability of Sweetpotato Anthocyanins by Impact of pH and Temperature

Purified PSPAs were derived from “Fresh”, “Dried” and “SD-w” samples, and prepared into series of pH solutions with the same concentration of initial PSPA (Figure 4A). Patterns for the three samples were similar, and data of “SD-w” were shown for the spectra (Figure 4B) and degradation (Figure 4C,D and Appendix A; Table 2). 

For freshly prepared PSPA solutions, the color (Figure 4A) and spectra (Figure 4B) were shown. From pH 1 to 7, the color gradually faded from magenta to pink, corresponding to decreased peak values and a gradual bathochromic shift of λmax from 521 nm to 545 nm (Table 2). The discontinuous color and spectra for pH 7, 8, and 9 were caused by the buffer systems, with pH 1.0 to 7.0 as citrate-phosphate buffers, pH 8.0 as a phosphate buffer, and pH 9.0 to 13.0 as glycine-NaOH buffers. Increasing pH from 9 to 13 resulted in a color change from pink to blue/green and then yellow, indicating an alternation in anthocyanin structure from the red flavylium cation to the blue/green quinonoid base and then yellow chalcones as reported [11,13].

The color and absorption spectra of PSPA solutions were performed at pasteurization temperature (65 °C) for 12 h (Figure 4C and Appendix A) and at 25 °C for 32 days (Figure 4D and Appendix A) to assess thermal degradation kinetics parameters (Table 2). The blue/green hues at pH 10–12 quickly turned yellow, while the magenta hue at lower pH displayed better color stability. PSPA exhibited extremely high color stability at pH 1–2 without obvious spectral decay at 65 °C and 25 °C.

In contrast to the gradual bathochromic shift from pH 1–13, fixed λmax wavelengths were displayed for solutions of each specific pH on the spectra collected over time (Appendix A). The degradation rate constant (k) and half-life time (t_1/2_) were calculated based on the peak values (Table 2). During the experiment, neither color nor spectrum decay was observed for PSPA solutions at pH 1 or 2 at either 65 °C or 25 °C. These results indicated effective storage of PSPA at pH 1–2, consistent with the high stability of PSPA in solvent (1) “1% HCl-methanol” (pH around 0) and solvent (4) “1% citric acid-ddH_2_O” (pH around 2.43) within 54 days’ storage (Figure 3A, Table 1).

When pH ≥ 3, degradation occurred, with accelerated speed at higher pHs and higher temperature indicated by higher k values and shorter t_1/2_. PSPA solutions were unstable under alkaline conditions, and the blue/green hues at pH 10–12 quickly turned yellow. PSPA at pH 3 exhibited t_1/2_ of 1.60 day at 65 °C and 29.04 day at 25 °C, whereas at pH 12 exhibited t_1/2_ of 0.21 day at 65 °C and 0.44 day at 25 °C. PSPA solutions at pH 13 were green when freshly prepared but turned yellow in minutes (not shown), and no anthocyanin peaks were identified. According to the declining absorption curves for pH 13 (Appendix A), the yellow color of the solutions faded during the experimental period (Figure 4C,D).

## 4. Discussion

### 4.1. Pre-Treatments by Steaming of Unpeeled Whole Sweetpotato Can Minimize Browning

Due to the high PPO content in sweetpotato flesh, the theoretically stable acylated anthocyanins can be vulnerable to rapid browning during processing [12,16,20]. This was demonstrated by the instant browning of fresh samples after being exposed to air. Moreover, PPO and many other plant enzymes are optimally active near 40–50 °C [16]. The effect of steaming pre-treatment on properties of PSPA varies in published reports [8,17,18,19,20]: With shredded or sliced sweetpotato, Kim et al. [8] reported nearly 50% PSPA yield reduction by steaming, Jiang et al. [17] displayed serious melanism of PSPA during storage, Liu et al. [18] reported browning issues, while good retention of PSPA were also reported [19,20]. Here, serious browning appeared for the cut surfaces of steamed samples of SD-s and SD-p, possibly caused by the strengthened browning reactions in the gradual heating process. The above inconsistent conclusions by steaming pre-treatment can be explained by different browning degrees relating to cutting surfaces. In contrast, SD-w samples displayed vibrant purple color both after steaming and as dried powder, with low brown indices.

The pre-treatment of “SD-w” by steaming of unpeeled whole sweetpotato has four advantages. (**1**) **Oxygen Excluding:** Protected by the peel, the flesh of whole sweetpotato would not be exposed to oxygen and is protected from O_2_-dependent browning of PPO. (**2**) **Enzyme inactivation:** PPO, glycosidase, and other oxidase enzymes naturally occur in plant tissues, contributing to the instability of anthocyanins [6,15,16]. The hot steam heating deactivates various enzymes in the flesh, preventing their influence at later steps and benefiting color stability of PSPA. This is also indicated in some other reports [12,16,20]. (**3**) **Avoidance of Maillard reaction:** The Maillard reaction easily takes place between reducing sugar and protein in a dry heat condition and causes non-enzymatic browning. However, it would not occur within wet steaming [30]. This is illustrated by the brown color of water extracts of “Dried” samples and the clear pink color for that of “SD-w” samples, consistent with other reports [8,19,30]. (**4**) **Tissue softening of the sweetpotato flesh:** The solid raw flesh turned soft after steaming, making it much easier to handle. Although an oven drying procedure was adopted in our study, it is suggested to immediately extract the soft flesh with solvent, as drying consumes additional energy and the obtained hard particles need to be powdered by a blender. Drying, however, may be preferred in case of long-term storage or if colorant flours are intended for sale.

### 4.2. Citric Acid-ddH_2_O as the Optimal Extraction Solvent for PSPA

Microwave- and ultrasonic-assisted extraction methods significantly improve phytochemical extraction efficiency, but the choice of solvents determines the quality of the extracts [21,31,32]. Acidified methanol is commonly used in the laboratory [5,32], and showed the highest yields of PSPA with high stability in our study, but it is inedible. Although acidified ethanol can be consumed, it displayed low extraction efficiency and quickly degraded PSPA within it. With pure water (ddH_2_O), the extraction efficiency was low and serious browning occurred for sweetpotato slices. However, the addition of citric acid greatly improves the extraction efficiency of water on PSPA—solvent (4) “1% citric acid-ddH_2_O” showed comparable extraction and preservation results as acidified methanol.

Besides the acidification function, the PPO inhibitory ability of citric acid also plays a significant role [16]. Citric acid is the most accessible and inexpensive among many PPO inhibitors (e.g., citric acid, oxalic acid, sodium borate, etc.), making “citric acid-ddH_2_O” the ideal edible solvent that can be applied to the food industry. Additionally, citric acid can be further employed to adjust the pH of the extracts to 1–2 for long-term storage.

### 4.3. Use of PSPA as Potential Natural Colorants 

Due to potential health risks that synthetic pigments may pose to humans, natural plant pigments have become increasingly favored in the food industry [1,2,3]. Many colorful foods are made by anthocyanin-rich fruits and vegetables, but it is difficult for them to maintain color consistency. Both internal and external factors would affect the color stability of anthocyanins [6,11,12,14].

Internal factors mainly include anthocyanin structures and the enzymes within plant tissues. The degree of acylation is positively correlated with anthocyanin stability [11,13]. However, the theoretically stable acylated PSPA can be easily browned or degraded by enzymes within raw sweetpotato [16,17]. This problem can be bypassed by steaming the whole pulp to inactivate related enzymes in an oxygen-free environment. Further inhibition of the PPO can be achieved by the addition of citric acid to the extraction solvents [16], as applied in “1% citric acid-ddH_2_O”.

Among the external factors, heat and pH conditions are essential during food processing. The importance of pH to anthocyanins stability is shown to be much more significant than temperature or light [22]. Anthocyanins are very susceptible to alkaline environments and are usually most stable at pH < 3 [11]. Our assays showed that PSPA solutions stored at 25 °C for 32 days or pasteurized at 65 °C for 12 h did not exhibit any obvious decay at pH 1–2, and a small decay at pH 3. Alkaline conditions could induce rapid degradation of PSPA, making them unsuitable for use in the food industry. Similar results were obtained with the PSPA stored at 20 °C for 35 days in research by He et al. [15]. According to the results, when at pH < 3, PSPA solutions can tolerate high disinfection temperatures during processing and long-term storage at room temperature, endowing PSPA with great potential as an ideal food colorant.

## 5. Conclusions

This study showed browning happened instantly when sweetpotato flesh was exposed to air, which markedly influenced the yield, color, and antioxidant capacity of PSPA. A steaming pre-treatment of unpeeled whole sweetpotatoes effectively prevented browning and improved the color stability. The solvent “1% citric acid-ddH_2_O” can be used as a valid edible solvent for PSPA extraction and preservation. Remarkably high stability was displayed for PSPA at pH 1–2. Alkaline conditions should be avoided when PSPA is applied. This work aligns with the growing trend towards natural and clean-label products that have increased the demand for natural colorants. The results can serve as valid guidance for manufacturers when incorporating anthocyanins into food products to ensure the desired color stability.

## Figures and Tables

**Figure 1 foods-13-00833-f001:**
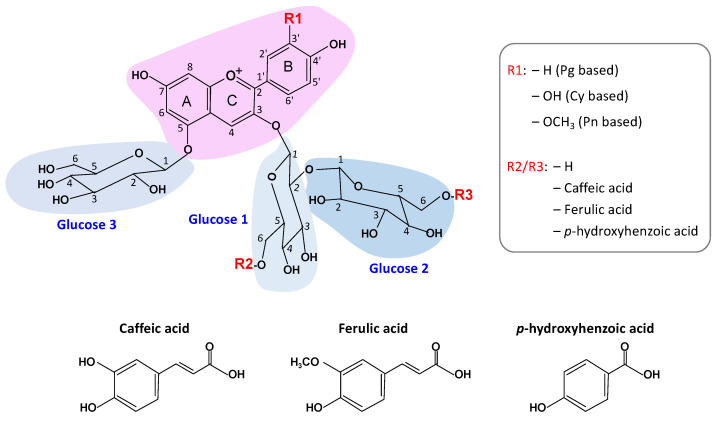
Structure of anthocyanins from purple sweetpotato. The pink shade indicates the anthocyanidin structure, the blue shade shows glycosylation by three glucose molecules, and the R2 and R3 are the positions for acylation. Modified from [6,8].

**Figure 2 foods-13-00833-f002:**
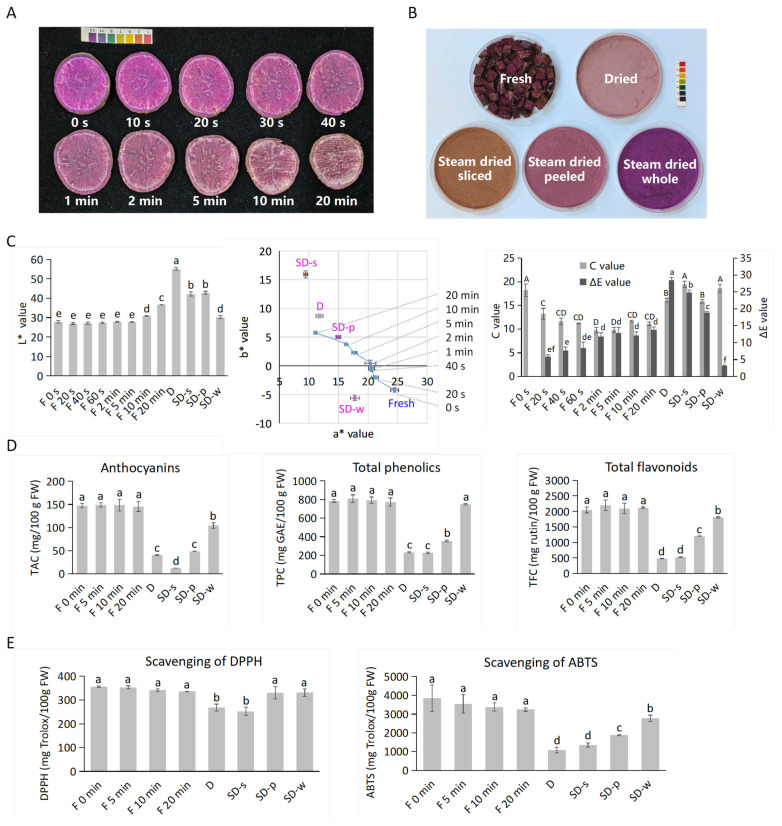
Effects of pre-treatments on the anthocyanin-related properties for purple sweetpotato. (**A**) the color change from 0 s–20 min for newly cut sweetpotato slices; (**B**) the powder color; and the (**C**) CIEL*a*b* parameters, (**D**) contents of anthocyanins, total phenolics, and total flavonoids, and (**E**) antioxidant activities by scavenging of DPPH and ABTS of fresh slices and powder from four pre-treatments. From (**C**–**E**), F is for Fresh, D is for Dried, and SD is for Steamed and Dried with varied pre-treatments. ΔE were between “F 0 s” and other samples. The universal pH indicator paper was shown as color reference. All values are based on three biological repeats. The difference significances were done by one-way ANOVA and Duncan’s test at *p* < 0.05, and the uppercase letters are used to show differences for C, while lowercase letters are used for ΔE and other indices.

**Figure 3 foods-13-00833-f003:**
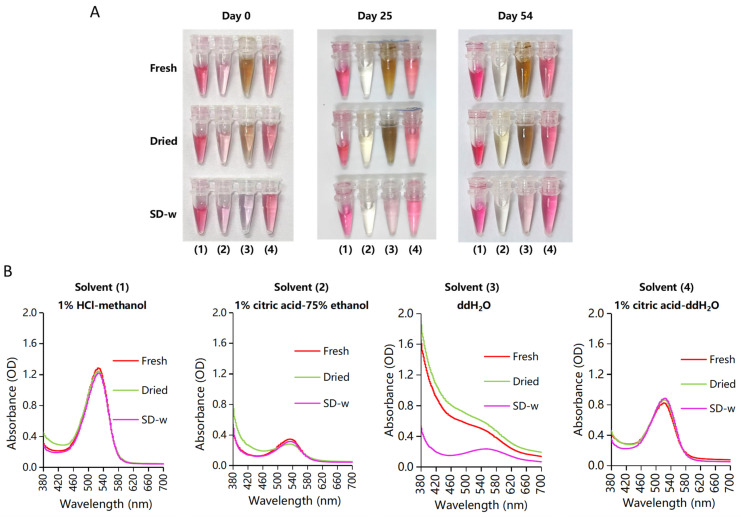
Color and spectra for the diluted extracts by four solvents with theoretically identical initial PSPA concentration. (**A**) Color of PSPA solutions at 0–54 days’ storage in the dark under room temperature. For photographs of Day 0 and 25, plain white printing paper was used as the background. For Day 54, glossy white paper (generally used for packaging boxes) was selected, resulting in clearer and brighter photos. (**B**) Absorption spectra of PSPA solutions at Day 0.

**Figure 4 foods-13-00833-f004:**
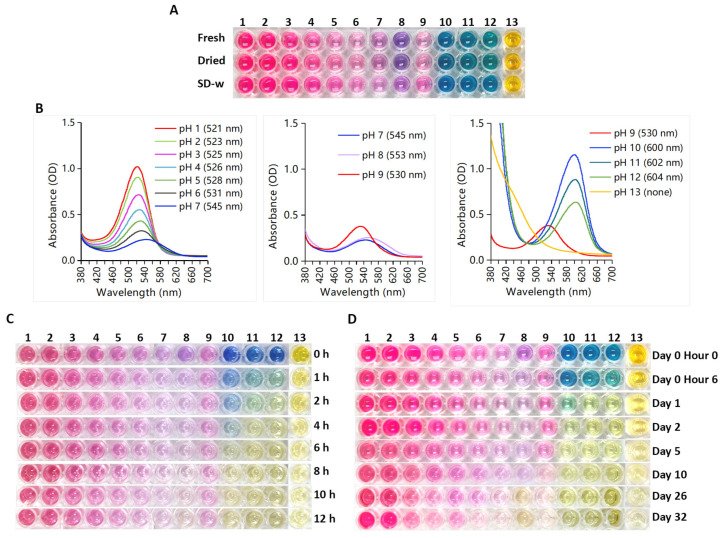
The joint impact of pH and temperature on the stability of PSPA solutions. (**A**) Color and (**B**) spectra of PSPA solutions at pH 1 to 13. The spectra of SD-w sample were shown and λmax for pH solutions were indicated in the brackets. Color change of PSPA solutions in dark at (**C**) 65 °C for 0–12 h, and at (**D**) 25 °C for 0–32 days.

**Table 1 foods-13-00833-t001:** The anthocyanin-related properties for purple sweetpotato extracts by four solvents.

	Pre-Treatment	Solvent (1)	Solvent (2)	Solvent (3)	Solvent (4)
pH of initial extracts	Fresh	0.54 ± 0.020	3.84 ± 0.003	6.18 ± 0.009	2.45 ± 0.009
	Dried	−0.32 ± 0.000	3.97 ± 0.003	5.90 ± 0.017	2.43 ± 0.003
	SD-w	−0.27 ± 0.003	3.96 ± 0.015	5.99 ± 0.003	2.43 ± 0.007
λmax (nm)	Fresh	526	532	–	520
	Dried	526	532	–	520
	SD-w	526	532	554	524
Anthocyanin yield (mg/100 g FW)	Fresh	115.92 ± 4.22 ^a^	61.14 ± 5.18 ^c^	10.62 ± 0.25 ^e^	80.48 ± 0.69 ^b^
	Dried	40.96 ± 1.04 ^d^	34.43 ± 1.63 ^d^	11.18 ± 0.61 ^e^	35.10 ± 0.25 ^d^
	SD-w	112.31 ± 9.58 ^a^	79.19 ± 2.72 ^b^	74.97 ± 1.19 ^b^	107.54 ± 1.08 ^a^
Polymeric color percent (%)	Fresh	17.5 ± 0.3 ^g^	17.9 ± 1.0 ^g^	99.8 ± 1.0 ^a^	23.2 ± 1.5 ^f^
	Dried	36.6 ± 1.4 ^d^	41.4 ± 0.3 ^c^	94.1 ± 1.5 ^b^	19.6 ± 0.7 ^g^
	SD-w	23.7 ± 0.1 ^f^	35.9 ± 0.9 ^d^	30.7 ± 1.2 ^e^	6.8 ± 0.3 ^h^
Brown index	Fresh	1.46 ± 0.02 ^h^	1.86 ± 0.07 ^d^	7.86 ± 0.08 ^a^	1.82 ± 0.01 ^d,e^
	Dried	1.63 ± 0.02 ^g^	2.00 ± 0.04 ^c^	2.33 ± 0.03 ^b^	1.73 ± 0.02 ^e,f^
	SD-w	1.37 ± 0.02 ^h^	1.62 ± 0.01 ^g^	1.67 ± 0.05 ^f,g^	1.26 ± 0.04 ^i^

The ‘–’ indicates that no absorption peak was identified. Significant differences were calculated using one-way ANOVA and Duncan’s test at *p* < 0.05, and the superscript letters reflect the data differences within each index.

**Table 2 foods-13-00833-t002:** Thermal degradation kinetic parameters for PSPA solutions at different pHs.

pH	λmax (nm)	65 °C			25 °C	
		k (h^−1^)	t_1/2_ (h)	t_1/2_ (day)	k (day^−1^)	t_1/2_ (day)
pH 1	521	~0	–	–	~0	–
pH 2	523	~0	–	–	~0	–
pH 3	525	0.018	38.51	1.60	0.024	29.04
pH 4	526	0.028	25.16	1.05	0.045	15.44
pH 5	528	0.036	19.29	0.80	0.058	11.95
pH 6	531	0.034	20.21	0.84	0.053	13.12
pH 7	545	0.049	14.14	0.59	0.066	10.50
pH 8	553	0.071	9.81	0.41	0.114	6.06
pH 9	530	0.038	18.30	0.76	0.058	11.90
pH 10	600	0.229	3.03	0.13	1.479	0.47
pH 11	602	0.180	3.85	0.16	1.766	0.39
pH 12	604	0.138	5.02	0.21	1.537	0.45
pH 13	none	–	~0	~0	–	~0

The ‘–’ means values unable to be determined. For PSPA solutions at pH 13 with peaks that vanished instantly, t_1/2_ ≈ 0 and k as “–”. For PSPA solutions at pH 1 and 2 with no decay of peak values during storage, k ≈ 0 and t_1/2_ as “–”. For PSPA solutions at 65 °C, values of t_1/2_ were provided both in hour and day for easier data comparison.

## Data Availability

The original contributions presented in the study are included in the article/Appendix A, further inquiries can be directed to the corresponding author.

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
