# Peer review of "Pre-Treatment, Extraction Solvent, and Color Stability of Anthocyanins from Purple Sweetpotato"

_foods, 2024, doi:10.3390/foods13060833_

Round 1
Reviewer 1 Report
Comments and Suggestions for Authors
The authors decided to evaluated the impact of pH and temperature on the properties of the Purple sweetpotato anthocyanins (PSPA) solutions, in cluding color stability, absorption spectra, and thermal degradation kinetics during storage using the commercialized purple sweetpotato cultivar "Violet". The whole experiment was prepared and performed properly and met all the requirements of scientific experiment. It should be noted that the authors also took into account the kinetics of color changes in their study, which is particularly important in food design and preparations. The work is in line with the growing trend towards natural and clean-label products has increased the demand for natural colorants like anthocyanins. Consumers often prefer food and beverages with colors derived from natural sources. However, the stability of anthocyanins can be influenced by factors such as pH, light, heat, and the presence of certain chemicals. Manufacturers need to consider these factors when incorporating anthocyanins into food products to ensure the desired color stability. In this regard, I consider the issues undertaken in the study to be extremely important. In this regard, I consider the issues addressed in the publication to be extremely important.
The authors presented a detailed characterization of the changes over time and the pH function of PSPA color. The study also took into account the type of raw material from which the anthocyanins were extracted. As optimal results, the authors considered the pre-treatment of steaming of unpeeled whole sweet potato and the solvent 18 “1% citric acid-ddH2O.
The authors provide valid guidance for the extraction, preservation, and application of PSPA as an anthocyanin-based food colorant in the food industry which is in line with the global trends in the development of food technology. Anthocyanins serve as attractive natural alternatives to synthetic food colorants, providing both visual appeal and potential health benefits. Their use aligns with consumer preferences for natural and minimally processed ingredients in food products. I support this claiming and I consider this research to be of great value, especially taking into account the beneficial effect of PSPA on human health.
I recommend minor revision. All my remarks are listed below:
Line 89: The analysis of color changes is for this study and the stated purpose is the basis in this regard, so please complete the detailed conditions during color measurement (calibration, angle, observer and light source and etc.).
Author Response
Dear reviewer,
Thank you very much for your positive comments. We revised the manuscript as suggested.
The main changes are as follows:
- Figure 1 was refined to emphasis the position of R1, R2 and R3.
- The color data of C (chromacity) and â–³E(color difference) were added in Figure 2, and more details were added in the Methods .
- Two extra references mentioned by the reviewers were added.
- Other revisions suggested by reviewers, that are highlighted by red, and can be shown by“Track Changes” function.
Our responses to your comments are as below:
Point 1
Line 89: The analysis of color changes is for this study and the stated purpose is the basis in this regard, so please complete the detailed conditions during color measurement (calibration, angle, observer and light source and etc.).
Response 1
Details were added in “Materials and Methods” (Line 98-100). The colorimeter were calibrated by the built-in whiteboard on the cap, and measured by lighting method 45/0 (45° circular lighting, 0° reception). Light source for illumination is full spectrum LED (D65 ) with field view at 10°.
Reviewer 2 Report
Comments and Suggestions for Authors
The manuscript „Pre-treatment, Extraction Solvent, and Color Stability of Anthocyanins from Purple Sweetpotato “ by Zhuo Chen, Jian Wang, Yang Lu, Qiang Wu, Yi Liu, Yonghua Liu, Sunjeet Kuma, Guopeng Zhu and Zhixin Zhu documents the search of pre-treatment and optimal extraction solvent for purple sweetpotato to maintain better anthocyanin stability.
My notes and observations:
* Centrifugation speed should be specified in x g, not rpm.
* Colors parameters were measured by using a portable intel-91 ligent chromatic meter (LS170, LinShang technology Co., Ltd., Shenzhen, China). Is it a certified device? I have doubts whether the data measured by this device are accurate and can be published in a high-level scientific journal. Figure 2C where the a* and b* values are given is perhaps confusing. I think the submission is not very common. In addition, when it comes to color changes and stabilities, it is useful to provide colour saturation (C*), and the total colour diference (E*). There is also no comparison with literature data. Data on color measurements, even for anthocyanin from purple sweetpotato, is certainly sufficient.
* Figure 3 A represents color for the diluted extracts by four solvents with theoretically identical initial PSPA concentration at 0–54 days’ storage in the dark under room temperature. Why, after 54 days of storage, PSPA solutions using extraction solvents (1) and (4) are brighter than the initial ones. Does it correlate well with published results (stability studies)?
* Some parameters are given as /100 g FW. When using dried samples, is their weight equal to the amount of fresh product? 1 g of fresh sample after drying weighs 0.35 g? And this value is used for all calculations (anthocyanin content, phenolic content, etc.)? Is this an accurate ratio without errors? or perhaps existing errors can influence that additionally processed and dried samples have lower values (phenolics content, anthocyanins content, etc.)?
* There is a lack of discussion and comparison with literature data regarding antioxidant activity, anthocyanins, total phenolics, and total flavonoids contents. Also, the discussion and comparison of kinetic parameters of thermal degradation could be wider.
*Some publications have DOI numbers in the bibliography, others do not. It would be good to have it unified.
Author Response
Dear reviewer,
Thank you very much for your comments. We revised the manuscript as suggested.
The main changes are as follows:
- Figure 1 was refined to emphasis the position of R1, R2 and R3.
- The color data of C (chromacity) and â–³E(color difference) were added in Figure 2, and more details were added in the Methods .
- Two extra references mentioned by the reviewers were added.
- Other revisions suggested by reviewers, that are highlighted by red, and can be shown by“Track Changes” function.
Our responses to your comments are as below:
Point 1
* Centrifugation speed should be specified in x g, not rpm.
Response 1
Revised as suggested..
Point 2
Colors parameters were measured by using a portable intelligent chromatic meter (LS170, LinShang technology Co., Ltd., Shenzhen, China). Is it a certified device? I have doubts whether the data measured by this device are accurate and can be published in a high-level scientific journal. Figure 2C where the a* and b* values are given is perhaps confusing. I think the submission is not very common. In addition, when it comes to color changes and stabilities, it is useful to provide colour saturation (C*), and the total colour diference (E*). There is also no comparison with literature data. Data on color measurements, even for anthocyanin from purple sweetpotato, is certainly sufficient.
Response 2
The colorimeter has the calibration certificate issued by the South China National Metrology and Testing Center. The standard deviation of â–³E*ab of the instrument is within 0.03, ensuring good data repeatability. Built-in calibration whiteboard on the cap, and built-in compensation optical path assisted stable measurement.
L*, a*, and b* are the basic data of color, while C and â–³E (color difference value) are obtained based on them. â–³ E can quantitatively represent the degree of color change, but cannot reflect the direction of color change; C represents the chromacity of colors, but cannot indicate what color it is. Here, by a* and b*, the yellowish process and fading of red color were clearly showed by increased b* and decreased a*. To highlight the increase in yellow in blue yellow and the decrease in red in red green, it is necessary to visually display the basic data of a and b.
Also, we have also added the data of C and â–³ E (Figure 2C, and line 208-213). We have added a data comparison (line 204-205) in the results section.
Point 3
* Figure 3 A represents color for the diluted extracts by four solvents with theoretically identical initial PSPA concentration at 0–54 days’ storage in the dark under room temperature. Why, after 54 days of storage, PSPA solutions using extraction solvents (1) and (4) are brighter than the initial ones. Does it correlate well with published results (stability studies)?
Response 3
Thank you for your careful observation. This is due to the change of the white background used for photographing. We used the white A4 printing paper as the background for Day 0 and 25. While at Day 54, we accidentally found the white paper with plastic film layer on the surface (commonly used for the packaging box) makes the image more clear and vivid, possibly due to the reflective effect of the plastic. We add an explanation in the figure legend about this (line 268-270).
The results of stability were consistent with text description of several published results (eg. He et al. 2015; Chen et al. 2019). But the images were rarely displayed. Also, time span of our assay were longer, and could provide valid information for readers.
Point 4
* Some parameters are given as /100 g FW. When using dried samples, is their weight equal to the amount of fresh product? 1 g of fresh sample after drying weighs 0.35 g? And this value is used for all calculations (anthocyanin content, phenolic content, etc.)? Is this an accurate ratio without errors? or perhaps existing errors can influence that additionally processed and dried samples have lower values (phenolics content, anthocyanins content, etc.)?
Response 4
Three replicates were conducted, with dryness rate of 35.1% ± 0.65% (revised in line 89). Each sample was tested according to its own dryness rate (revised in line 107-108), and the method was described as “about 0.35g” for simplicity
Point 5
* There is a lack of discussion and comparison with literature data regarding antioxidant activity, anthocyanins, total phenolics, and total flavonoids contents. Also, the discussion and comparison of kinetic parameters of thermal degradation could be wider.
Response 5
As the “Results” and “Discussions” are separate in the manuscript, we did not conduct literature data comparison in the “Results”. The “Discussions” focuses on the theme of the subheadings, and missed data comparison for this part. In order not to affect the continuity of the discussion section, we have added a short data comparison in the results section (line230-232).
Point 6
*Some publications have DOI numbers in the bibliography, others do not. It would be good to have it unified.
Response 6
Revised as suggested. We added the DOI numbers for all references.
Reviewer 3 Report
Comments and Suggestions for Authors
The manuscript under review is well-written and structured. However, I have concerns about the novelty of the research, which is also not clearly articulated. The impact of pretreatment on various properties of sweet potatoes has been extensively reported in the literature. For instance:
- Li et al. (2019) discussed the extraction, identification, stability, bioactivity, application, and biotransformation of purple sweet potato anthocyanins.
- Mahmudatussa’adah et al. (2019) studied the effect of blanching pre-treatment on the colour and anthocyanin of dried slice purple sweet potato.
- Arindra (2021) investigated the effect of three blanching treatments on purple sweet potato anthocyanin extract retention.
- Ngamwonglumlert et al. (2017) reviewed natural colorants, focusing on pigment stability and extraction yield enhancement via the utilization of appropriate pretreatment and extraction methods.
Given these extensive studies, it is crucial to identify the gaps that this study aims to fill. While there might be some novelty, it has not been clearly addressed in the introduction. I acknowledge the information provided on Lines 65-71, but it seems more methodological than novel.
I would also suggest the authors consider using more accurate methods such as HPLC for the quantification of anthocyanins. If not possible, the reason needs to be given and justify the validity of using spectrophotometry instead.
In Table 1, the statistical analysis appears to be missing. This is a crucial aspect that needs to be addressed to ensure the validity of the results presented.
Lastly, the conclusion section requires a major revision. The current version reads more like an abstract and does not succinctly summarise the key findings and their implications.
I hope these comments are helpful in improving the manuscript.
Comments on the Quality of English Language
A proofreading for minor errors is required.
Author Response
Dear reviewer,
Thank you very much for your comments. We revised the manuscript as suggested.
The main changes are as follows:
- Figure 1 was refined to emphasis the position of R1, R2 and R3.
- The color data of C (chromacity) and â–³E(color difference) were added in Figure 2, and more details were added in the Methods .
- Two extra references mentioned by the reviewers were added.
- Other revisions suggested by reviewers, that are highlighted by red, and can be shown by“Track Changes” function.
Our responses to your comments are as below:
Point 1
*The manuscript under review is well-written and structured. However, I have concerns about the novelty of the research, which is also not clearly articulated. The impact of pretreatment on various properties of sweet potatoes has been extensively reported in the literature.
Given these extensive studies, it is crucial to identify the gaps that this study aims to fill. While there might be some novelty, it has not been clearly addressed in the introduction. I acknowledge the information provided on Lines 65-71, but it seems more methodological than novel.
Response 1
Revised as suggested (line 65-68,73-75).
The main novelty: (1) Instant browning of cutting surfaces was reported here, and explained some inconsistent conclusions of extraction efficiency by various pre-treatment. (2) Image display of PSPA solutions after long-term storage. (3) The kinetics of color changes were tointo account, at wider pH range (1-13) and longer time span.
Point 2
**I would also suggest the authors consider using more accurate methods such as HPLC for the quantification of anthocyanins. If not possible, the reason needs to be given and justify the validity of using spectrophotometry instead.
Response 2
HPLC-MS is a reliable and accurate method for measuring anthocyanins, but there is a standardized process for qualitative and quantitative analysis of anthocyanins. Fixed parameters are needed, including the extraction solvents (e.g. 70% ethanol, 5% formal acid, etc.), temperature, and pH. So, it is not very suitable for comparison of extracting solvents here.
The structures of PSPA characterized by HPLC-MS have been demonstrated in multiple studies. They basically conform to summary of Kim et al. (2012) (presented as Figure 1, Line 40-45).
Many study used HPLC-MS for initial characterization of PSPA in sweetpotato, but adopted pH differential method for subsequent comparison (eg. Jiang et al. 2019; ). The pH differential method is specialized for quantitative analysis of anthocyanins. Although it has multiple steps, it is widely applied in various solvent background such as beverages and fruit juices.
Point 3
*In Table 1, the statistical analysis appears to be missing. This is a crucial aspect that needs to be addressed to ensure the validity of the results presented.
Response 3
Revised as suggested. The statistical analysis for difference significances were added at the end of the figure legends. (Figure 2, Line 222; Table 1 Line 252-253)
Point 4
*Lastly, the conclusion section requires a major revision. The current version reads more like an abstract and does not succinctly summarise the key findings and their implications.
Response 4
Revised as suggested. (Line 349-403)
Point 5
*Comments on the Quality of English Language:A proofreading for minor errors is required.
Response 5
Thank you for your advise. We had our manuscript checked again by a native English-speaking colleague.